# Site-specific MCM sumoylation prevents genome rearrangements by controlling origin-bound MCM

Yun Quan[1], Qian-yi Zhang[1], Ann L. Zhou[1], Yuhao Wang[1], Jiaxi Cai[1], Yong-qi Gao[1], Huilin Zhou[1,2]*

1 Department of Cellular and Molecular Medicine, University of California at San Diego, La Jolla, CA, United States of America, 2 Moores Cancer Center, School of Medicine, University of California at San Diego, La Jolla, CA, United States of America

* huzhou@health.ucsd.edu

**Data Availability Statement:** All relevant data are within the manuscript and its Supporting Information files.

## Abstract

Timely completion of eukaryotic genome duplication requires coordinated DNA replication initiation at multiple origins. Replication begins with the loading of the Mini-Chromosome Maintenance (MCM) complex, proceeds by the activation of the Cdc45-MCM-GINS (CMG) helicase, and ends with CMG removal after chromosomes are fully replicated. Post-translational modifications on the MCM and associated factors ensure an orderly transit of these steps. Although the mechanisms of CMG activation and removal are partially understood, regulated MCM loading is not, leaving an incomplete understanding of how DNA replication begins. Here we describe a site-specific modification of Mcm3 by the Small Ubiquitin-like MOdifier (SUMO). Mutations that prevent this modification reduce the MCM loaded at replication origins and lower CMG levels, resulting in impaired cell growth, delayed chromosomal replication, and the accumulation of gross chromosomal rearrangements (GCRs). These findings demonstrate the existence of a SUMO-dependent regulation of origin-bound MCM and show that this pathway is needed to prevent genome rearrangements.

## Author summary

Faithful replication of the genome is essential for the survival and health of all living organisms. The eukaryotic genome presents a unique and difficult challenge: its enormous size demands the coordinated action of numerous DNA replication origins to ensure timely completion of genome duplication. Although the mechanisms that control the activation and removal of DNA replisome are partially understood, whether and how cells regulate the loading of the Mini-Chromosome Maintenance (MCM) complex, the precursor of the DNA replisome, at replication origins are not. Because mutations to MCM-loading factors and enzymes that catalyze reversible protein sumoylation cause substantial gross chromosomal rearrangements (GCRs) that characterize the cancer genome, understanding regulated MCM loading is one of the most pressing questions in the field. Here, we identified a site-specific SUMO modification of MCM and found that mutation

**Funding:** NIH RO1 GM116897 (H.Z.), S10 OD023498 (H.Z.) and University of California CRCC grant (HZ). The funders had no role in study design, data collection and analysis, decision to publish, or preparation of the manuscript.

**Competing interests:** The authors have declared that no competing interests exist.

disabling this modification causes severe growth defect and impaired DNA replication. These defects are attributable to reduced MCM at DNA replication origins, resulting in a lower DNA replisome level and a dramatic accumulation of GCRs. Thus, these findings identify a hitherto unknown regulatory mechanism: Site-specific MCM sumoylation regulates origin-bound MCM, and this prevents genome rearrangements.

## Introduction

Gross chromosomal rearrangements (GCRs) characterize many cancers [1] and include chromosomal translocation, inversion, interstitial deletion, and *de novo* telomere addition [2]. Most GCRs are consequences of inappropriate repair of DNA double-stranded breaks (DSBs) [3]. Numerous "at-risk" DNA sequences that exist in the eukaryotic genome affect how DSBs give rise to GCRs [4,5]. For instance, segmental duplication can mediate GCR formation via homologous recombination, and specific genetic pathways prevent these duplication-mediated GCRs (dGCRs) [6]. In particular, enzymes that catalyze reversible sumoylation, including the Mms21 SUMO E3 ligase, have a specific role in preventing dGCRs [7,8]. In the same vein, mutations affecting essential chromosomal DNA replication factors cause similar accumulations of dGCRs [9]. These parallel findings raise the possibility that SUMO may regulate DNA replication to prevent GCRs. Published findings supporting this idea are circumstantial and not by direct evidence [7,10,11]. The identification of a precise point of DNA replication control by SUMO is required to conclusively connect these processes.

Direct evidence that connects SUMO regulation of DNA replication and GCR suppression has been lacking for several reasons. First, eukaryotic DNA replication is a complex process that involves over 40 proteins in *Saccharomyces cerevisiae* and many more in vertebrates [12–15]. Most DNA replication proteins are essential for cell viability. Hypomorphic DNA replication mutants accumulate GCRs [9], but the non-specific nature of such mutants does not allow the identification of a precise point of DNA replication control by SUMO. Second, many DNA replication proteins are modified by SUMO [10,16]. Existing studies have not identified specific sumoylation sites or events that prevent the accumulations of GCRs. For instance, though sumoylation-deficient mutants of PCNA and DNA polymerase ε accumulate modest GCRs [17,18], SUMO E3 ligase mutants accumulate dGCRs at significantly higher rates [8], suggesting that sumoylation of other DNA replication proteins is crucial for genome maintenance. Prior studies suggested that SUMO modifications of MCM and Dbf4-dependent kinase (DDK) might inhibit DNA replication initiation [19,20]. However, the lack of site-specific sumoylation-deficient *mcm* and *dbf4* mutants in these studies prevents a rigorous test of their roles in suppressing dGCRs. The observation that a temperature-sensitive *cdc6* mutant accumulates higher levels of dGCRs than either temperature-sensitive *cdc7* or *dbf4* mutant suggests regulated MCM loading is more important than replication initiation in suppressing this type of GCR [9]. Nevertheless, direct evidence is lacking. Although many DNA replication proteins are modified by SUMO [11,16,19], the one(s) involved in GCR suppression has not been identified. Third, like other post-translational modifications, SUMO can act either in a site-specific manner or as a group modification [21–23]. These models make different predictions about the consequence of SUMO modification. The lack of known functional SUMO targets that suppress GCRs has made distinguishing between these two modes of SUMO regulation difficult. Here, we present evidence that a site-specific sumoylation of the Mcm3 subunit plays a critical role in preventing GCRs, and this MCM-SUMO modification regulates the MCM

loaded at DNA replication origins, providing the first evidence for regulated MCM loading in cells.

## Results

We set out to examine SUMO modification of Mcm3, a prominent and recurring SUMO substrate protein that emerged from our proteomic studies of cellular SUMOylation [10,11]. Inspection of the MCM double hexamer (MCM-DH, PDB: 5BK4 and 6F0L) structures identifies 39 surface-exposed lysines on *S. cerevisiae* Mcm3 [24,25]. To identify candidate SUMOylation sites, a panel of *mcm3-KR* mutants was generated that changed 8 to 39 lysines to arginine. Arginine cannot be modified by SUMO but retains key properties of the unmodified lysine side chain. The *mcm3-KR* mutants were tested for their abilities to complement the *mcm3Δ* mutant by plasmid shuffling (Fig 1A). Selection of cells without the complementing extragenic *MCM3* using 5-fluoroorotic acid (5-FOA) revealed that all *mcm3-KR* mutants grew poorly, including *mcm3-8KR*, which has the fewest KR mutations tested (Fig 1A). Faster growing colonies were occasionally observed, suggesting that these cells had acquired suppressor mutations. To identify these mutations, we plated ~1 million *mcm3-8KR* cells on 5-FOA and then purified the plasmids from faster growing, 5-FOA resistant and HIS+ colonies (Fig 1B). We found that the coding sequences for the five lysine residues between K759 and K784 of Mcm3 are restored to the wild-type sequence in suppressor alleles, converting *mcm3-8KR* to *mcm3-3KR* through gene conversion and indicating the 5KR mutations are responsible for the poor growth of the *mcm3-8KR* mutant.

K767 and K768 are conserved among fungal Mcm3 proteins, and this conservation extends to mouse and human Mcm3 (Fig 1C). Strikingly, we found that substituting both K767 and K768 to arginine, yielding *mcm3-2KR*, severely impairs cell growth (Fig 1D, top panel). A single *mcm3-K768R* mutant causes a modest but noticeable growth defect, while a single *mcm3-K767R* mutant does not (Fig 1D, top panel). Conversely, restoring either K767 or K768 in *mcm3-8KR* or *mcm3-39KR* rescues growth (Fig 1D, bottom panel), indicating the presence of either K767 or K768 is sufficient for robust growth.

Wild-type Mcm3 displays three distinct and slower migrating sumoylated species when analyzed by Western blot [11]. To identify SUMO-modified lysine residues, clusters of surface-exposed lysines on Mcm3 were changed to arginine (Fig 2A). Distinct Mcm3 sumoylated species are not affected by *smt3-allR*, in which all nine lysine residues of Smt3 are mutated to arginine to prevent poly-SUMO formation (Fig 2B), suggesting that mono-sumoylation of different lysine residues of Mcm3 contribute to the observed gel mobility shifts. We found that *mcm3-4KR* (lysine residues 352–357) and *mcm3-7KR* (lysine residues 624–639) eliminate the two slower migrating species, while *mcm3-14KR-1* (lysine residues between 681–714 and 784–967, plus K767R) or *mcm3-14KR-2* (lysine residues 681–714 and 784–967, plus K768R) drastically reduce the fastest migrating sumoylated Mcm3 species (Fig 2C). Among the lysine residues (K352, K355, K356, and K357) that are mutated in *mcm3-4KR*, a single K357R eliminates the slowest migrating sumoylated species to the same extent as *mcm3-4KR*, while K352R and K355R/K356R have little effect (S1 Fig), indicating that mono-sumoylation of K357 is responsible for this sumoylated Mcm3 species. Since SUMO-modified Mcm3 is a branched polypeptide during gel electrophoresis, this may contribute to the observed abnormal electrophoretic mobility (Fig 2B), consistent with the idea that SUMO modifications of lysine residues in different regions of Mcm3 are involved.

The fastest migrating sumoylated Mcm3 persists in *mcm3-14KR* mutants, suggesting the presence of additional SUMO modifications in the N- and C- termini of Mcm3. To determine whether K767 and K768 of Mcm3 are modified by SUMO, we next examine sumoylated

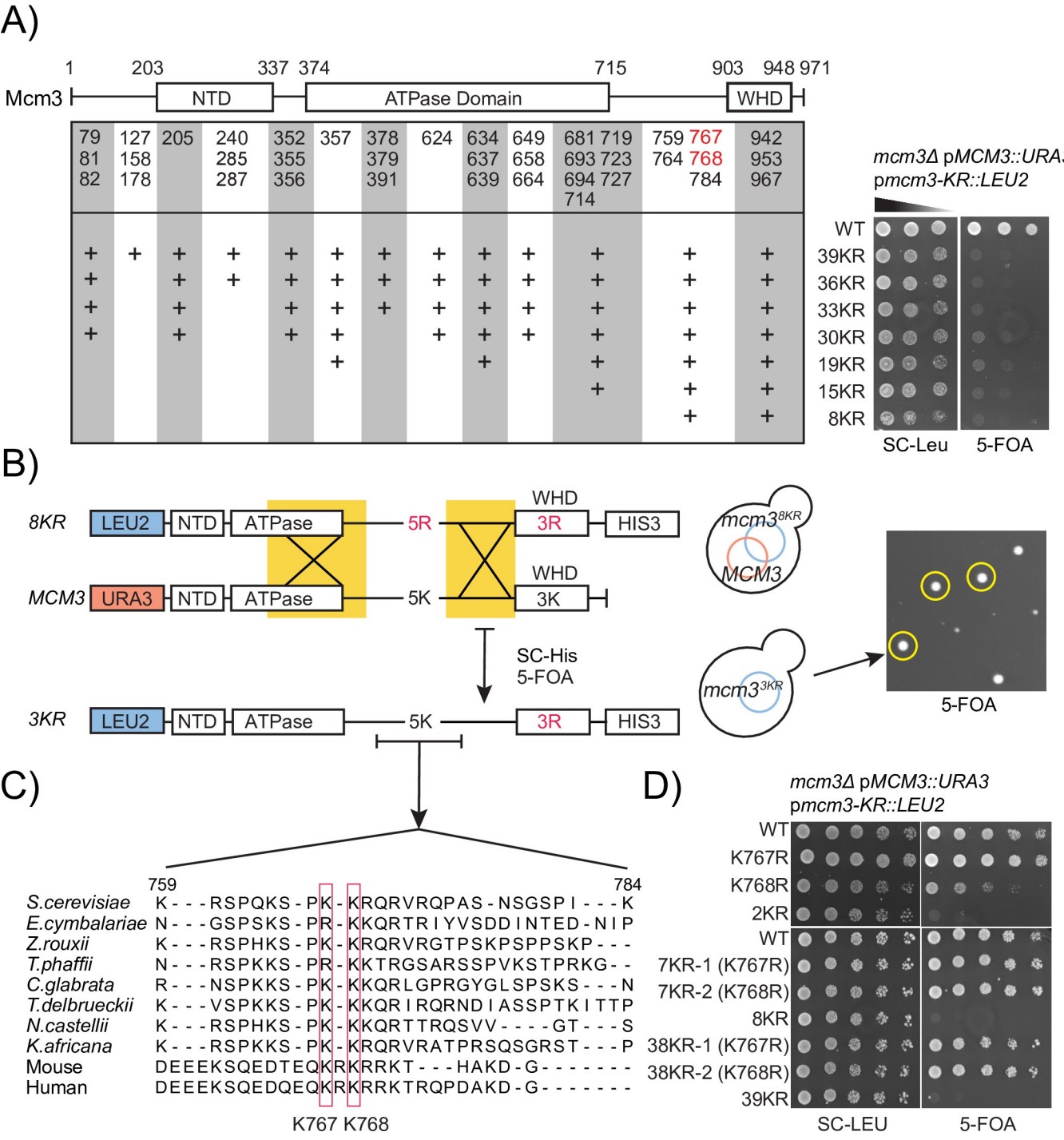

**Fig 1. Mcm3 K767/768R mutations impair cell growth.** (A) Left: schematic of surface-exposed lysines on Mcm3 (numbers correspond to amino acid positions) that were mutated into arginine, which cannot be modified by SUMO. Right: growth of serial dilutions of the various *mcm3-KR* mutants on media in which the complementing *MCM3* plasmid was retained (SC-Leu) or lost (5-FOA). (B) Gene conversion between the *MCM3* and *mcm3-8KR* mutant plasmids led to faster growing 5-FOA resistant and HIS+ colonies in which the lysines between K759-K784 of Mcm3 were restored to the wild-type sequence. (C) Sequence alignment of fungal, mouse, and human Mcm3 shows that K767 and K768 of *S. cerevisiae* Mcm3 are conserved. (D) An intact K767 or K768 in Mcm3 is necessary and sufficient to restore growth. Top: effects of mutating K767R, K768R, or both on cell growth on 5-FOA plates. In this case, no other *mcm3* mutations exist. Bottom: effects of restoring K767 or K768 in *mcm3-8KR* and *mcm3-39KR* backgrounds on cell growth on 5-FOA plates. *mcm3-7KR-1* and *mcm3-38KR-1* have K767R and K768, while *mcm3-7KR-2* and *mcm3-38KR-2* have K767 and K768R.

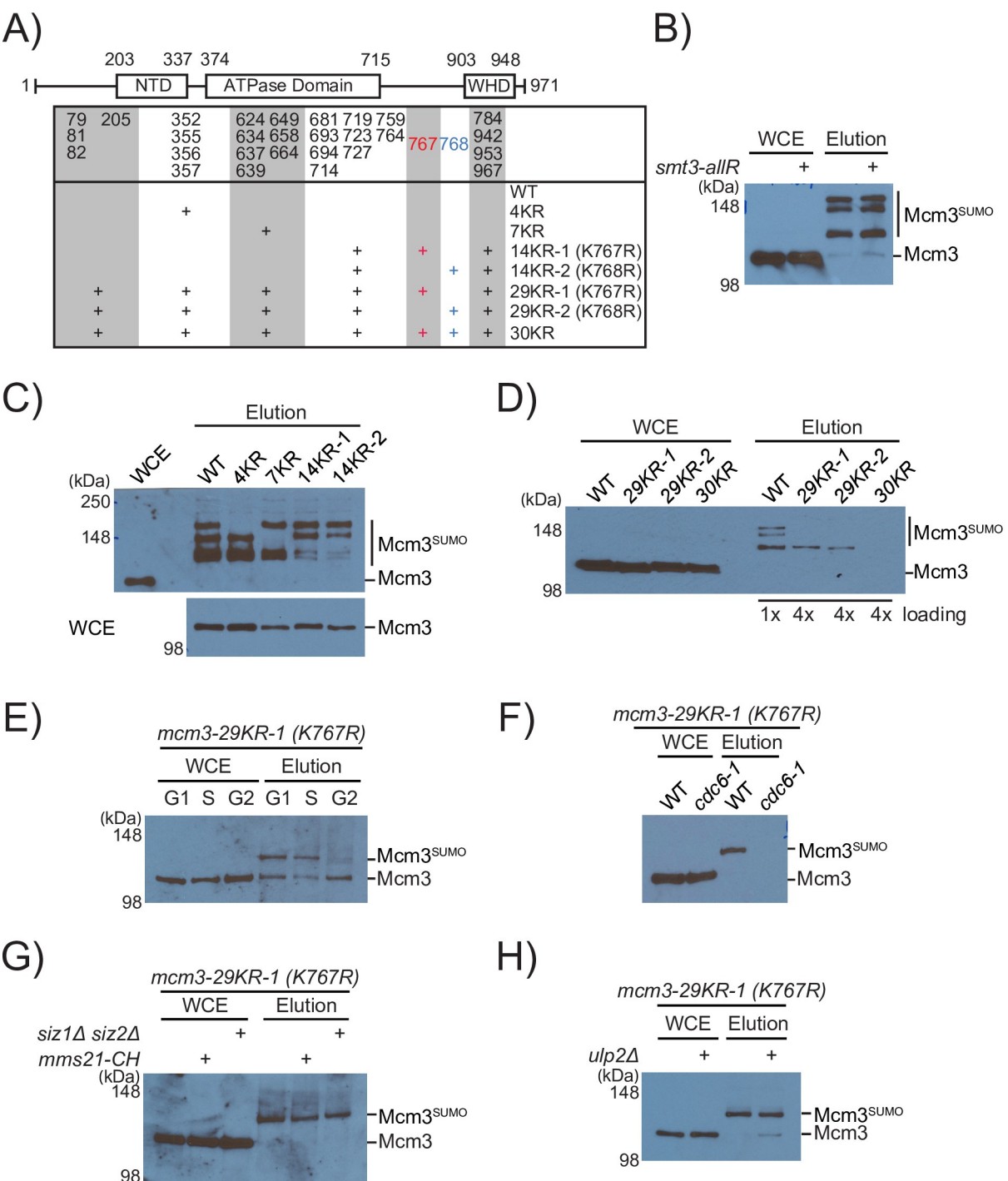

**Fig 2. K767 and K768 of chromosome-bound Mcm3 are sumoylated.** (A) Schematic of the *mcm3-KR* mutants analyzed. Total sumoylated proteins in each mutant were purified using Ni-NTA and anti-Flag affinity purifications and analyzed by the anti-Mcm3 antibody (see Methods). (B) Removal of poly-sumoylation by *smt3-allR* does not affect sumoylated species of Mcm3. (C) Specific clusters of KR mutations (also see Fig 2A) in Mcm3 eliminated distinct sumoylated species, indicating the specificity of its SUMO modifications. The effect of mcm3-4KR has been attributable to K357R (S1 Fig). (D) A single sumoylated species in either *mcm3-29KR* mutant (contains K767R or K768R) is completely eliminated by *mcm3-30KR*. 4-fold more loading of the eluted samples were used for the *mcm3-29KR/30KR* mutants compared to WT, indicating that K767 and K768 sumoylation represents a small percentage of Mcm3 sumoylated species. (E) Sumoylated K768 in the *mcm3-29KR-1 (K767R, K768)* mutant fluctuates during the cell cycle, being highest in G1, slightly lower in S and absent in G2/M phase cells. FACS confirms the expected cell cycle stages (S1 Fig). (F) Inactivating Cdc6 in G1 eliminated K768 sumoylation in the *mcm3-29KR-1 (K767R)* strain. *mcm3-29KR-1 (K767R)* strains carrying *CDC6* or *cdc6-1* were arrested in G1 by alpha-factor for 4 hours at 37°C. FACS confirms

G1-arrested cells (S1 Fig). (G) Effect of mutating SUMO E3 ligases (*siz1Δ*, *siz2Δ* and *mms21-CH*) on sumoylated K768 of Mcm3 in the *mcm3-29KR-1 (K767R)* strain. (H) *ulp2Δ* does not affect sumoylated K768 of Mcm3 in the *mcm3-29KR-1 (K767R)* strain.

Mcm3 in WT, *mcm3-29KR-1* (29 surface-exposed lysines, including K767R), *mcm3-29KR-2* (29 surface-exposed lysines, including K768R), and *mcm3-30KR* (30 surface-exposed lysines) cells. Both *mcm3-29KR* mutants eliminate the majority of sumoylated Mcm3, except for the fastest migrating species, which is eliminated by *mcm3-30KR*, in which both K767 and K768 were mutated to arginine (Fig 2D). Thus, SUMO modification of K767 and K768 is responsible for this sumoylated Mcm3 species, which represents a small percentage of its overall sumoylation (Fig 2D, 4X loading of the mutants), but it has a more important role in maintaining cell growth than the other lysine residues (Fig 1).

Next, we used the *mcm3-29KR-1 (K767R)* mutant to evaluate the regulation of K768 sumoylation. Like the bulk sumoylated Mcm3 [11,19], K768 sumoylation peaks in G1, declines in S and disappears in G2/M (Fig 2E), mirroring the cell cycle behavior of chromosome-bound MCM complexes [26]. Cdc6 is an essential factor required for MCM-loading in G1 [27,28]. Inactivating Cdc6 by shifting the *cdc6-1* mutant to non-permissive temperature (37˚C) in G1 eliminates sumoylated K768 in the *mcm3-29KR-1 (K767R)* background (Fig 2F), like the bulk sumoylated MCM [19]. Thus, K768 sumoylation occurs on the origin-bound Mcm3. We found that K768 sumoylation is subjected to redundant control by three SUMO E3 ligases (Fig 2G). The Ulp2 desumoylase prevents the formation of poly-SUMO chain but has a non-uniform and subunit-specific effect in suppressing MCM sumoylation [11,29,30]. Mutations removing Ulp2 or poly-SUMO chain do not appreciably affect dGCRs [11]. We found that Ulp2 loss does not noticeably affect sumoylated K768 of Mcm3 (Fig 2H). Thus, this site-specific SUMO modification of Mcm3 is not subjected to poly-sumoylation.

Upon integration at the chromosomal *MCM3* locus, the single K767R or K768R mutant does not affect cell growth or the levels of sumoylated Mcm3 in asynchronous cells. Interestingly, *mcm3-2KR* (K767R and K768R) drastically slows cell growth and unexpectedly reduces all three sumoylated species of Mcm3 (Fig 3A). Moreover, G1-arrested *mcm3-2KR* cells display lower levels of sumoylated Mcm2 and Mcm6 (Fig 3B), indicating that *mcm3-2KR* indirectly reduces sumoylation of other MCM subunits. Because SUMO modifies origin-bound MCM [19], this indirect effect suggests that *mcm3-2KR* cells may have a lower amount of origin-bound MCM, resulting in an overall reduced level of sumoylated MCM. Indeed, ChIP-qPCR analysis shows that *mcm3-2KR* drastically reduces the association of Flag-Mcm3 at an early origin (ARS305) in G1 cells (Fig 3C). Similarly, *mcm3-2KR* reduces origin-association of Mcm4 and Mcm6, the latter at both early and late origins (Fig 3D and 3E). Thus, *mcm3-2KR* reduces the amounts of MCM loaded at both early and late origins, although this analysis does not exclude the possibility of uneven reductions of origin-bound MCM elsewhere in the genome.

MCM loading involves the formation of MCM single hexamer (MCM-SH) and MCM double hexamer (MCM-DH) [31,32]. G1 cells have the maximal amount of loaded MCM complex, while G2/M cells have the least. To detect the endogenous MCM complex, we immuno-purified Flag-Mcm3 from G1 and G2/M cells. Considerably more Mcm2, Mcm6 and Mcm7 co-purify with Flag-Mcm3 in G1 cells than in G2/M cells, indicating that this assay detects primarily the functional MCM complex (Figs 4A and S2A). Interestingly, *mcm3-2KR* reduces the amount of co-purified MCM subunits in G1 cells, suggesting a reduction of this MCM complex, in agreement with the reduced level of origin-bound MCM detected above (Fig 3). By contrast, *mcm3-2KR* has no detectable effect on the association among MCM subunits in G2/M cells that lack the loaded MCM. Thus, *mcm3-2KR* specifically affects the MCM complex in

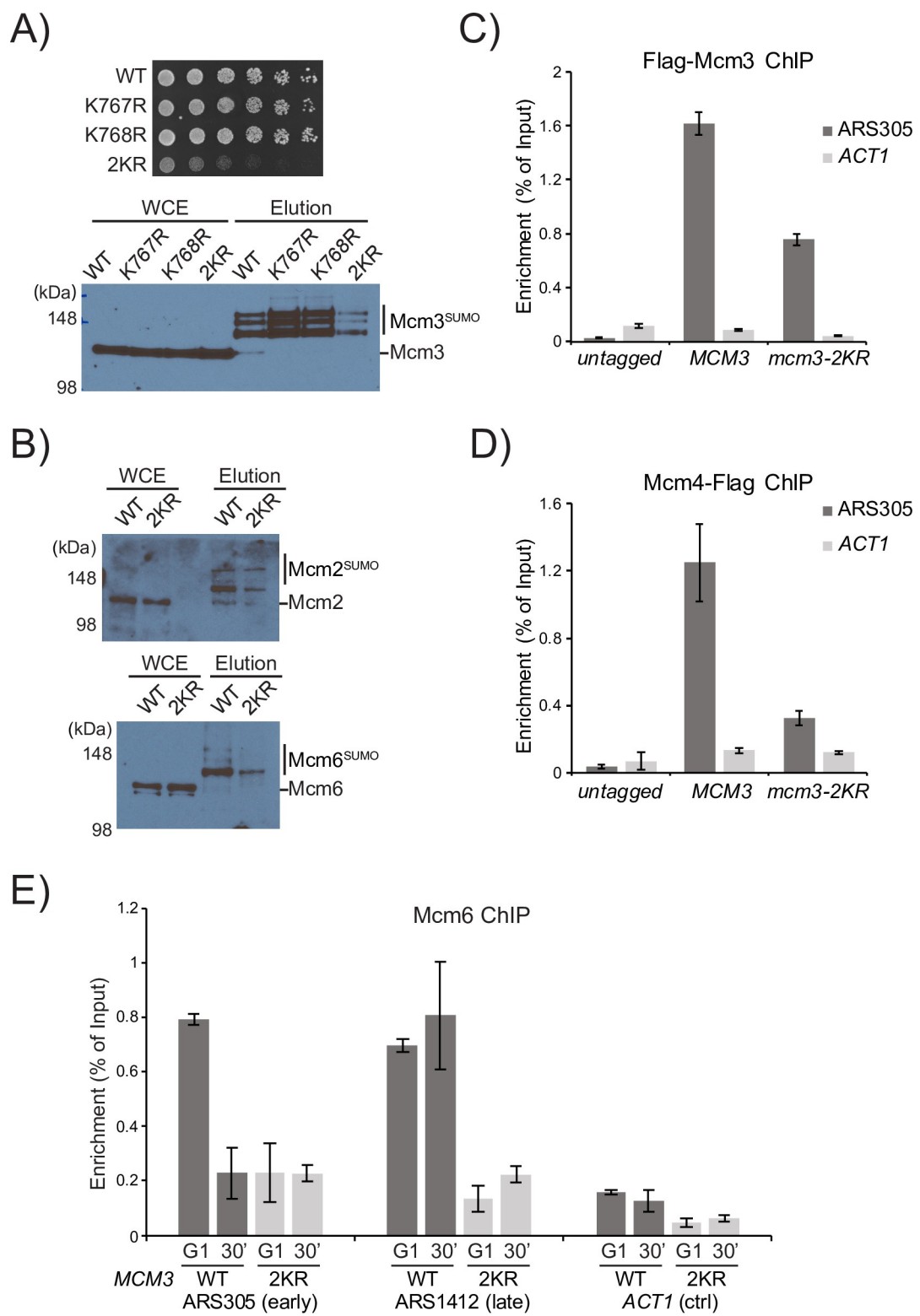

**Fig 3. *mcm3-2KR* reduces MCM sumoylation and impairs its loading at both early and late origins.** (A) The *mcm3-2KR* (K767R and K768R) mutant grows poorly and has a significantly reduced overall sumoylation, while K767R or K768R has little effect on cell growth or bulk Mcm3 sumoylation. These *mcm3* mutants were integrated at the chromosomal locus. (B) *mcm3-2KR* reduces sumoylated Mcm2 and Mcm6 in alpha-factor arrested G1 cells, showing that *mcm3-2KR* indirectly reduces sumoylation of the other MCM subunits. (C-D) ChIP-qPCR of Flag-Mcm3 and Mcm4-Flag in *mcm3-2KR* cells revealed

reduced MCM at the origin ARS305 in G1. (E) ChIP-qPCR showed that *mcm3-2KR* reduced Mcm6 associations at both early origin (ARS305) and late origin (ARS1412) in G1 and S phase cells.

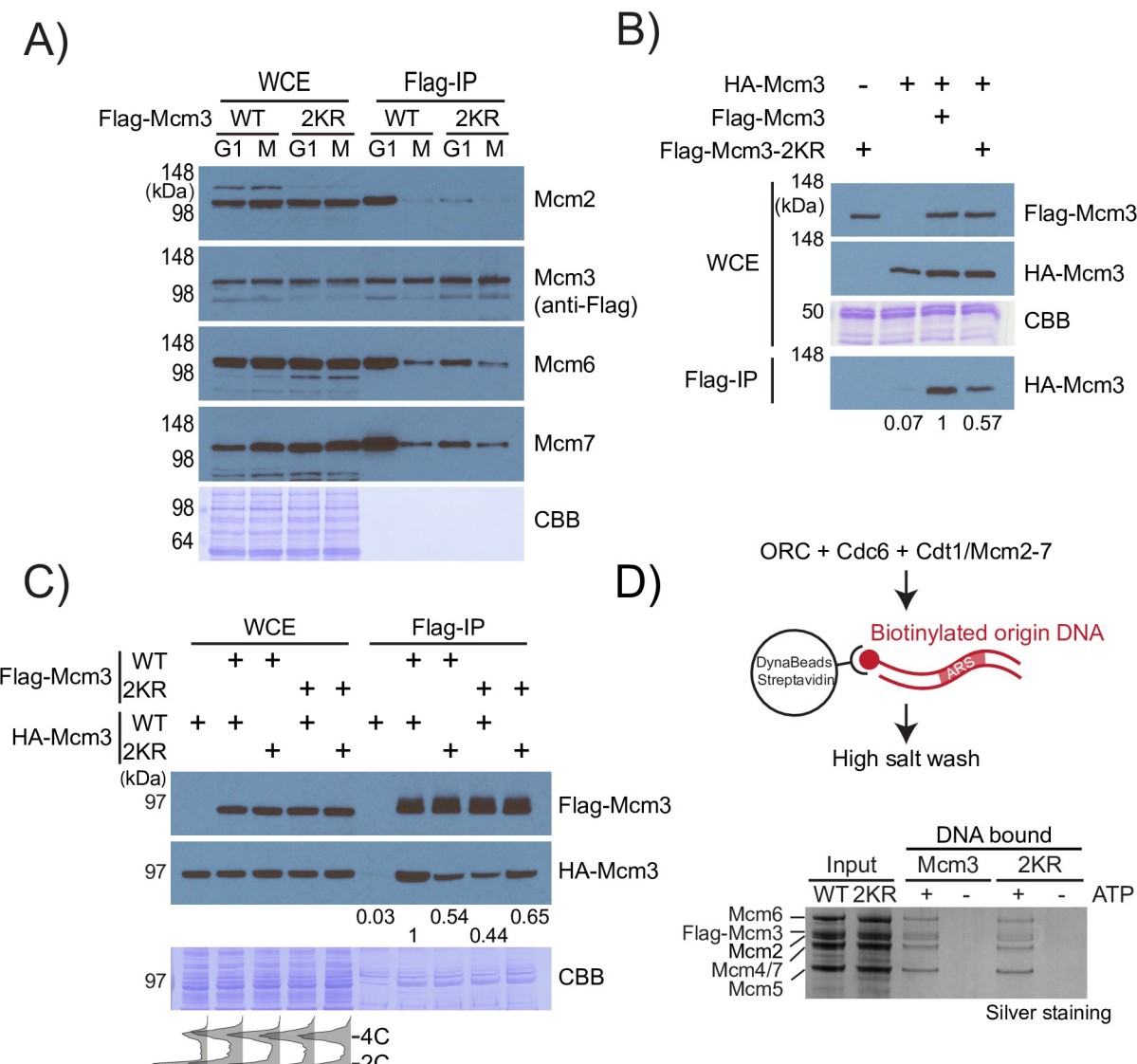

**Fig 4. *mcm3-2KR* reduces MCM complex formation *in vivo*, but not *in vitro*.** (A) Effects of *mcm3-2KR* and cell cycle stages on the association among MCM subunits. Flag-Mcm3 from the indicated cells was immuno-purified using anti-Flag affinity resins and probed by various anti-MCM antibodies. (B) G1 arrested HA-Mcm3 haploids expressing extragenic WT Flag-Mcm3 or Flag-Mcm3-2KR were used for anti-Flag immunoprecipitation. Co-purified HA-Mcm3 was only observed when both tagged Mcm3 proteins were present, confirming the specificity of this assay in detecting MCM-DH. A lower level of co-purified HA-Mcm3 was found to associate with Flag-Mcm3-2KR. Densitometry was used to quantify the Mcm3 protein bands; the relative intensity to the most intense band is shown. (C) Diploid cells expressing Flag or HA tagged Mcm3, WT or 2KR, were used for anti-Flag immunoprecipitation assay. Co-purified HA-Mcm3 is maximal in WT diploids, lowest in *MCM3/mcm3-2KR* heterozygous diploids and intermediate in homozygous *mcm3-2KR* diploids. Thus, wild-type Mcm3 is preferentially used to make MCM-DH, and such preference is not available in the homozygous *mcm3-2KR* diploid. CBB: Coomassie blue staining. The numbers below the anti-HA Western blot indicate the relative amount of HA-Mcm3 based on densitometric analysis. (D) *In vitro* MCM-DH loading using immobilized origin-containing DNA template, Cdc6, ORC and MCM-Cdt1, which contains either WT or 2KR mutant of Flag-Mcm3. Following high salt wash, the loaded MCM-DH was eluted and detected by silver staining. The requirement of ATP confirmed MCM-DH loading, showing that *mcm3-2KR* does not affect MCM-DH loading *in vitro*.

G1 without affecting MCM complex formation in G2/M. A caveat of this assay is that it does not specifically detect MCM-DH, the final product of MCM loading. To specifically detect MCM-DH, we employed a dual tagging strategy that distinguishes MCM-DH from other MCM sub-complexes [33]. We found that less HA-Mcm3 co-purifies with Flag-Mcm3-2KR than with Flag-Mcm3 WT in G1-arrested haploids, suggesting a reduction in the level of MCM-DH that includes two copies of the Mcm3 subunit (Figs 4B and S2B). Using diploid strains expressing chromosomal Flag/HA-Mcm3 in various combinations, we found that heterozygous *mcm3-2KR/MCM3* diploids have lower levels of HA-Mcm3 that co-purified with Flag-Mcm3, irrespective of the placement of the 2KR mutation (Figs 4C and S2C). In each of these pull-down experiments, slower migrating sumoylated species of MCM subunits were not detected, indicating that their stoichiometry of sumoylation is low.

To determine if reduced MCM-DH levels in *mcm3-2KR* cells are due to a defect in forming MCM-DH, we loaded MCM-DH on an origin-containing DNA template using purified Cdc6, ORC and the MCM-Cdt1 complex that contains either Flag-Mcm3 or Flag-Mcm3-2KR. We found that *mcm3-2KR* has no detectable effect on the purified MCM-Cdt1 complex or the loaded MCM-DH *in vitro* (Fig 4D), ruling out an intrinsic defect of Mcm3-2KR in forming MCM-DH *in vitro*. Therefore, SUMO controls MCM-DH formation *in vivo* but not *in vitro*, indicating the presence of an extra SUMO-dependent step in controlling the amount of MCM loaded in cells, which has yet to be included here.

What are the consequences of inactivating the SUMO-dependent step in controlling the level of MCM loaded in cells? Compared to wild-type cells, *mcm3-2KR* cells display delayed S phase entry and slower progression through S phase (Fig 5A). Psf2 is a subunit of GINS that forms the CMG helicase with Cdc45 and MCM [34,35]. CMG, detected as Mcm3 that co-precipitates with Psf2, appears more slowly at the onset of S phase and persists longer in *mcm3-2KR* cells (Fig 5B). At its peak, CMG levels are noticeably lower in *mcm3-2KR* than in wild-type cells. Titration analysis confirms that *mcm3-2KR* cells have approximately 50% less fully assembled CMG than wild-type cells (Fig 5C). This 2-fold reduction of the CMG level implies that chromosomal replication could take at least twice as long to complete, in agreement with the slower growth of *mcm3-2KR* cells (Fig 2A).

The slower growth of *mcm3-2KR* cells is not caused by aberrant hyper-activation of the DNA damage checkpoint, which is characterized by hyper-phosphorylated Rad53 kinase [36]. Hyper-phosphorylated Rad53 is undetectable in unperturbed *mcm3-2KR* cells (Fig 5D), indicating that spontaneous DNA damage or replication stress is below the detection limit. Hydroxyurea (HU) treatment causes stalled DNA replication forks, which trigger Rad53 hyper-activation via the Mrc1 adaptor in the DNA replisome [37–40]. HU-induced (or Phleomycin-induced) Rad53 activation is dampened in *mcm3-2KR* cells compared to WT cells, consistent with the idea that Rad53 activation requires signaling via an intact DNA replisome, and that replisome levels are lowered in the *mcm3-KR* cells. These findings show that *mcm3-2KR* cells progress through the cell cycle slowly because of insufficient CMG helicase, although they do not accumulate high enough levels of spontaneous DNA damage or replisome stress to cause appreciable cell cycle arrest.

Pulsed-field gel electrophoresis (PFGE) shows that DNA replication and initiation are delayed in *mcm3-2KR* cells. While wild-type cells complete chromosome replication within 45 minutes of entering S phase, discrete (fully replicated) chromosomes in *mcm3-2KR* cells does not fully disappear at the beginning of S phase, nor do they completely return even after 120 minutes after the start of S phase (Fig 6A), indicating delays in both initiation and completion of chromosomal replication. The sluggishness of chromosome replication in *mcm3-2KR* cells suggests that incomplete DNA replication could occur. This could cause DSBs to form GCRs. Consistent with this idea, the dGCR rate is elevated 9-fold in *mcm3-K767R*, ~100-fold in

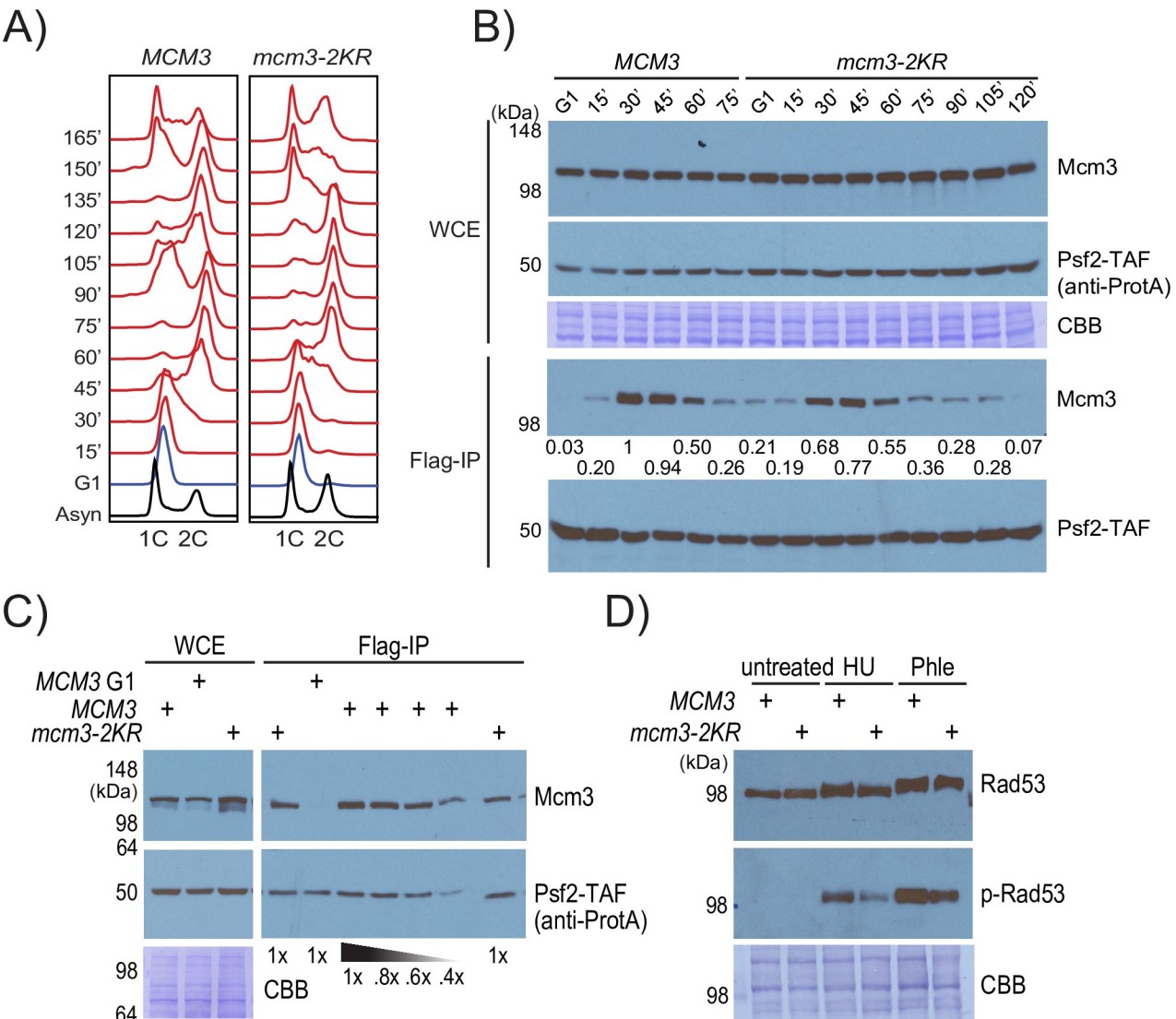

**Fig 5. Effects of *mcm3-2KR*: slower cell cycle progression and lower CMG levels.** (A) Fluorescence-Activated Cell Sorting (FACS) analysis of WT and *mcm3-2KR* mutant following G1-arrest and release into the cell cycle. (B) The CMG level during the cell cycle was measured by the amount of Mcm3 that co-purified with Psf2, a subunit of GINS. Psf2 was detected by anti-ProteinA antibody, while Mcm3 was detected by anti-Mcm3 antibody. The relative amount of co-purified Mcm3 was quantified by densitometry, normalized against the most intense signal in the same blot. (C) CMG levels in *mcm3-2KR* cells were reduced to ~50% compared to that of WT cells. The eluted sample (WT) was titrated (1x, 0.8x, 0.6x and 0.4x) relative to the fixed loading (1x) for the *mcm3-2KR* mutant. (D) Hyper-phosphorylated Rad53 was not detectable in unperturbed *mcm3-2KR* cells, but those induced by HU or Phleomycin was dampened. Top panel: total Rad53 level was detected by anti-Rad53 antibody. Middle panel: antibody specific to phosphorylated Rad53 was used to detect the activated form of Rad53. Bottom panel: Coomassie blue (CBB) staining of the cell lysates used. Cells were treated by 200 mM HU or 50 μg/ml phleomycin for 2 hours.

*mcm3-K768R*, and over 200-fold in *mcm3-2KR* (Fig 6B and S1 Table). This is comparable to the rates of dGCRs accumulation in the SUMO E3 ligase-null mutants [7,8] and MCM loading mutants such as *cdc6-1*. In contrast, DNA replication initiation mutants such as *cdc7* and *dbf4* have significantly lower dGCR rates [9], suggesting that the level of MCM loaded at origins plays a more important role in suppressing dGCR than the origin-firing activity. Although *mcm3-2KR* has a modestly higher rate (~ 2-fold) of dGCRs than that of *mcm3-K768R*, it has a severe growth defect (~ 2-fold slower). However, we caution against making a quantitative correlation between these distinct phenotypes, considering the GCR assay measures events near

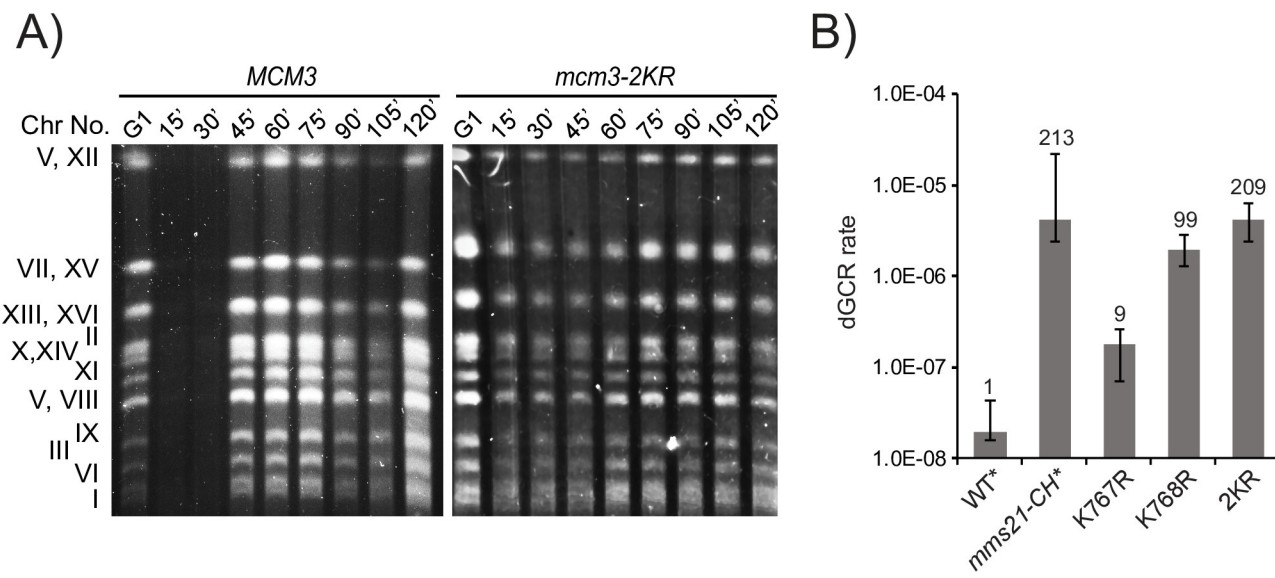

**Fig 6. *mcm3-2KR* revealed reduced chromosomal replication, and drastic accumulation of dGCRs.** (A) PFGE analysis of yeast chromosomes in WT and *mcm3-2KR* cells released from G1-arrest. Distinct chromosome bands correspond to fully replicated chromosomes. (B) Rates of accumulating dGCRs in *mcm3* mutants, showing the fold-increase caused by K767R, K768R, and 2KR mutations; dGCR rates of WT and *mms21-CH* were taken from a previous study for comparison [8].

the telomere on the left arm of Chromosome V, while cell growth is a collective response of all origin activities.

## Discussion

The assembly and activities of distinct MCM complexes on chromosomes define the step-by-step process of eukaryotic DNA replication [15,41]. The identification of site-specific SUMO modification of Mcm3 here, together with the analysis of these MCM complexes, uncovers a hitherto unknown function of SUMO in regulating DNA replication through the control of origin-bound MCM levels. To prevent re-replication, MCM is loaded once per cell division and this is restricted to G1. This restriction means that there must be a sufficient supply of loaded MCM-DH that can be activated to form CMG, a limiting activity that ensures the timely completion of genome replication. Indeed, a modest 2-fold reduction in MCM-DH and CMG levels in the *mcm3-2KR* mutant causes a drastically impaired cell growth, severely reduced DNA replication, and over 100-fold increase in dGCRs, supporting the idea that MCM-DH and CMG levels are the limiting factors for eukaryotic DNA replication. Although the regulatory mechanisms that control CMG activation and removal have been studied [15,42,43], regulatory mechanisms that control the loaded MCM levels have not. The findings here suggest that site-specific Mcm3 sumoylation controls this critical step. Besides Mcm3, SUMO also modifies other MCM subunits and other DNA replication proteins [11,16,19]. Their SUMO modifications may also contribute to the control of MCM loading and/or other steps in DNA replication. Our findings here indicate that a conclusive demonstration of their specific involvements in GCR suppression should wait for the identification of site-specific sumoylation-deficient mutants.

The time needed to replicate eukaryotic genome depends on the amount of origin-bound MCM, which controls the level of CMG helicase that unwinds DNA, the rate-limiting step of DNA replication. How might SUMO modification of Mcm3 ensure optimal MCM-DH levels in cells? We propose two possibilities. First, Mcm3 initiates MCM loading via its C-terminal

winged helix domain (WHD) [44]. Sumoylation sites (K767 and K768) in the vicinity of this WHD suggest that they may work together to promote MCM loading either directly or indirectly. Existing findings show that MCM loading occurs in a step-by-step process that involves MCM-SH as an intermediate and MCM-DH as the final product [31,32,45]; each of these steps could be subjected to the control by this SUMO modification. Although the co-immunoprecipitation experiment does not reveal the precise nature of the MCM complex (Fig 4A), *mcm3-2KR* specifically reduces its level in G1 when MCM loading is maximal, and it has no detectable effect on the MCM complex formation in G2/M when the loaded MCM is absent. Notably, *mcm3-2KR* has no detectable effect on the expression of Mcm3 protein or that of any other MCM subunits in cells, suggesting that this mutant disrupts a specific regulation of the G1-specific MCM complex. Determining the precise step(s) regulated by this SUMO modification would require a complete reconstitution of SUMO-dependent MCM loading. To this end, our initial attempts to reconstitute SUMO-dependent MCM-DH loading *in vitro* are inconclusive, pointing to the possibility that other unknown cellular factors may be needed. Second, SUMO has been shown to trigger protein degradation or complex disassembly via poly-SUMO and the ubiquitin system [46,47]. Contrary to this prevailing model, our findings here suggest that SUMO modification of Mcm3 at K768 may inhibit MCM unloading. We found no detectable poly-SUMO chain emanating from sumoylated K768, even after Ulp2 removal (Fig 2). Mutations affecting Ulp2 or poly-SUMO also do not appreciably affect dGCRs [11], ruling out a major role of poly-sumoylation in this process [46,48]. An earlier study showed that Cdc6 is necessary for the establishment and maintenance of the pre-replicative complex [27], suggesting that MCM unloading could occur. If so, this site-specific SUMO modification might prevent this putative MCM unloading activity through unknown mechanisms. Further studies are needed to test these proposals.

The findings here show that regulated MCM loading is critical for maintaining genome stability. MCM sumoylation is a key part of the regulatory mechanism. Mutations disrupting site-specific Mcm3 sumoylation cause the MCM-DH and CMG levels to fall, delaying both the initiation and completion of chromosomal DNA replication. Though the source of spontaneous DSBs that accumulate in the *mcm3* mutants has yet to be identified, incomplete DNA replication is a possibility and is particularly dangerous to cells, since DSBs that occur in the un-replicated region cannot be repaired via homologous recombination with a sister chromatid. Such DSBs must search for partially homologous sequences elsewhere in the genome to initiate break-induced replication via non-allelic recombination, a process that causes dGCRs [3,6]. We observed precisely these genomic insults in the *mcm3* mutants. Similar dGCRs have also been observed in mutants affecting MCM loading factors and enzymes in the SUMO pathway [7–9]. Our biochemical findings connect these genetic observations and suggest that regulated MCM loading through site-specific Mcm3 sumoylation prevents incomplete DNA replication and consequently genome rearrangements.

## Material and methods

### Yeast genetics method and plasmid construction

Standard yeast genetics methods were used to construct the strains used in this study (S2 Table). DNA sequencing of the genomic DNA confirmed point mutations integrated at the chromosomal locus. Unless noted otherwise, plasmids were constructed using DNA recombination repair and confirmed by DNA sequencing (S3 Table). Details of these constructions are available upon request.

## Method to detect sumoylated MCM following tandem affinity purification of SUMOylated proteins

For each purification, approximately 200 mL of *HF-SUMO* cells at an $OD_{600}$ of 1 were collected and washed with 20 mM iodoacetamide in PBS buffer and then aliquoted into four screw-cap tubes. To each tube, 500 μL of glass beads and 800 μL of acid-lysis solution (1.25% SDS/0.25 M HCl) were added, and then the tubes were vortexed at maximum speed for 10 minutes at room temperature. The lysate was neutralized with 1 M NaOH and pH was further adjusted to 8 with 100 μL $NaPO_4$ (pH 8.0). After adding 10 mM DTT, the lysates were heated at 95˚C for 10 minutes. Following centrifugation, the clarified cell lysates of each strain were combined, to which 30 mM Iodoacetamide was added to quench the DTT. Protein concentrations of different samples were normalized, and each lysate (3.5 mL) was incubated with 200 μL packed Ni-NTA beads (Bio-Rad) for 2 hours at room temperature. The beads were washed sequentially with 6 mL of PBSN (PBS with 0.2% NP-40) and 2 mL of PBSN/0.1% SDS. For elution, 400 μL Elution buffer-I (PBSN, 0.1% SDS, 25 mM EDTA) and 800 μL Elution buffer-II (PBSN, 25mM EDTA with protease inhibitors) were sequentially added to the Ni-NTA beads and the eluted samples were combined. The Ni-eluted samples were then incubated with 20 μL anti-FLAG M2 beads (Sigma) for 2 hours at room temperature. The anti-FLAG resins were washed with 4 x 1 mL PBSN and then incubated with 60 μL of 1x LDS-buffer (Invitrogen) at 60˚C for 5 minutes. The anti-FLAG eluted samples, separated from the anti-FLAG resins, were heated at 95˚C in the presence of 10 mM DTT for 10 minutes. 10 μL of 1/200 dilution of the cell lysate was used as the input sample, while 30 μL of the eluted samples were analyzed by SDS-PAGE using 8% acrylamide gels. The gel was then transferred to a PVDF membrane, which was subsequently blocked with 5% milk at 4˚C overnight. Polyclonal anti-Mcm3, anti-Mcm2, and anti-Mcm6 antibodies were affinity-purified from rabbit sera (Covance), using the corresponding MCM proteins immobilized on CNBr resins.

## ChIP-qPCR to detect MCM-loading at origins

To evaluate the localization of Mcm3, 4 and 6 to replication origins, ChIP was performed [49,50]. Briefly, yeast cultures (150 mL, for three immunoprecipitation experiments) were grown to an $OD_{600}$ of 0.8 and cross-linked for 15 min with 1% formaldehyde at room temperature. Whole-cell lysates were prepared in 1.4 mL of ChIP lysis buffer (50 mM Hepes, pH 7.6; 140 mM NaCl; 1 mM EDTA; 1% Triton, and 0.1% sodium deoxycholate) supplemented with protein inhibitors (2 mM phenylmethylsulfonyl fluoride, 200 μM benzamidine, 0.5 μg/mL leupeptin, 1 μg/mL pepstatin A, 1 mM PMSF) by glass beads beating and sonication to shear the genomic DNA to an average size of 300–500 bp. Immunoprecipitation was performed using 50 μL Dynabeads ProteinG (ThermoFisher) and 2.5 μL anti-Flag antibody M2 (Sigma; F3165), or 6 μg anti-Mcm6 antibody. After binding, beads were washed as follows: once in 1 mL lysis buffer with 5 min incubation, twice in 1 mL washing buffer (100 mM Tris-Cl, pH 8.0; 250 mM LiCl; 0.5% NP-40; 0.5% deoxycholate; and 1 mM EDTA) with 5 min incubations, and once with 1 mL TE buffer (10 mM Tris-Cl, pH 8.0, and 1 mM EDTA) with 1 min incubation. After the washes, the samples were first eluted in 40 μL of TE buffer with 1% SDS at 65˚C for 10 min, and this was saved as elution 1. 162 μL of DNA extraction buffer (135 μL of TE, 15 μL of 10% SDS, 12 μL of 5 M NaCl) was then added to the beads with 1.5 μL of RNaseA and incubated at 37˚C for 30min, and this was elution 2. Both eluates were mixed and incubated at 37˚C for another 30 min. The inputs were treated with 162 μL of DNA extraction buffer and 1.5 μL of RNaseA with one hour incubation at 37˚C.

The input and immunoprecipitated DNA were incubated in the same buffer with the addition of 20 μg Protease K at 65˚C overnight to reverse crosslinks before purification using a

QIAquick PCR Purification kit (QIAGEN). Before qPCR analysis, the input DNA was diluted 1:100, and immuno-purified DNA was diluted 1:10 by volume. qPCR was done using SYBR Green 2x master mix (KAPA Biosystems) on a Roche LightCycler 480 system. Three independent immunoprecipitation experiments were performed. qPCR primer sequences were:

ARS305_fwd: TCATGTACTGTCCGGTGTGA
ARS305_rev: CCGTTTTTAGCCCCCGTGTA
ARS1214_fwd: ACTGACCGCGGCTAAAAGTT
ARS1214_rev: CTCCTCTCTACTTGCGTGTGT
ACT1_fwd: AGAGTTGCCCCAGAAGAACA
ACT1_rev: GGCTTGGATGGAAACGTAGA

### *In vivo* MCM haxamer assay using co-immunoprecipitation

To evaluate the formation of MCM complex, haploid cells expressing N-terminal tagged 3xFlag-Mcm3 or 3xFlag-Mcm3-2KR were used to purify Mcm3 via anti-Flag affinity resins (M2, Sigma). Briefly, 100 mL of cells at $OD_{600}$ of 0.8, arrested in G1 by alpha factor (15 ng/mL) or G2/M by nocodazole (15 μg/mL), respectively, were harvested by centrifugation. Cell pellets were washed once with 1 mL PBS buffer (1.06 mM $KH_2PO_4$, 5.6 mM $K_2HPO_4$, 154 mM NaCl, 10% Glycerol, pH 7.4) supplemented with 20 mM Iodoacetamide and N-Ethylmaleimide before being frozen at -80˚C. Cell lysates were prepared by glass beads beating in 1 mL of lysis buffer (50 mM HEPES-K pH 7.5, 150 mM K-Glutamate, 2.5 mM $Mg(OAc)_2$, 0.1 mM $ZnCl_2$, 3 mM ATP, 0.2% NP-40, protease inhibitors) supplemented with 60 μg DNaseI (Grade II, Roche), followed by clarification by centrifugation on a benchtop centrifuge at 4˚C. 900 μL of cell lysates were incubated with 20 μL of anti-Flag M2 agarose beads for 3 hours at 4˚C, followed by washes with 1 mL lysis buffer for 5 times. Bound fractions were eluted by incubating the beads in 80 μL of 1.5xLDS sample buffer at 65˚C. Anti-Flag, anti-Mcm2, anti-Mcm6, and anti-Mcm7 Western blots were performed to detect various MCM subunits. MCM antibodies were produced in rabbits immunized with recombinant MCM proteins (Covance), followed by affinity purification using recombinant MCM antigens.

### *In vivo* MCM-DH assay using dual tagging strategy

To evaluate the levels of MCM double hexamer, haploid cells expressing 6xHis-3xHA-Mcm3 from its endogenous locus and 3xFlag-Mcm3 from a single-copy plasmid under the control of *MCM3* promoter, or diploid cells expressing both 6xHis-3xHA-Mcm3 and 3xFlag-Mcm3 were used. Haploid cells were grown in 50 mL of SC-Leu to an $OD_{600}$ of 0.4 and arrested in G1 phase by the addition of 15 nM alpha factor for 3 hours; diploid cells were grown in 50 mL of YPD to an $OD_{600}$ of 1 and then harvested by centrifugation. Cell pellets were washed once with PBS (1.06 mM $KH_2PO4$, 5.6 mM $K_2HPO4$, 154 mM NaCl, 10% Glycerol, 0.2% NP-40, pH 7.4) supplemented with 20 mM Iodoacetamide and N-Ethylmaleimide before frozen at –80˚C. Cell lysates were prepared by glass beads beating in 0.8 mL of PBS supplemented with 0.2% NP-40 (PBSN), 60 μg DNaseI and protease inhibitors, followed by clarification by centrifugation on a benchtop centrifuge at 4˚C. Cell lysates were normalized to the same concentration and incubated with 20 μL of anti-Flag M2 agarose beads for 3 hours at 4˚C, followed by washes with 1 mL PBSN for 4 times. Bound fractions were eluted by incubating the beads in 25 μL of 2xLDS sample buffer at 65˚C. Anti-Flag and anti-HA (Sigma, clone 3F10) Western blots were performed to detect 3xFlag-Mcm3 and 6xHis-3xHA-Mcm3, respectively, in the whole cell lysates and bound fractions. Densitometric analysis of Western blots regarding the co-purified 6xHis-3xHA-Mcm3 was performed using ImageJ. Briefly, Western blot films were scanned and converted into grayscale pictures. In ImageJ, a frame was drawn around the

relevant band to define the Region of Interest; the mean gray value of each band and background were measured. The pixel density for all data was used for the calculation of the relative amount of each band, normalized against the most intense band in the same blot.

## MCM-Cdt1 purification

The MCM-Cdt1 complex was purifed as described previously [41,44]. The MCM-Cdt1 cells expressing either WT Flag-Mcm3 (yJF38) or Flag-Mcm3-2KR (HZY2301) were grown in 1 L of YP-Raffinose to an $OD_{600} = 0.8$ and arrested in G1 with 50 nM alpha factor for 3 hours. MCM-Cdt1 expression was induced with 2% galactose for 5 hours. The cells were harvested by centrifugation, washed with Buffer A (25 mM HEPES-KOH pH 7.6, 5 mM Mg(OAc)$_2$, 0.1 M K-Glutamate, 0.02% NP-40, 10% Glycerol), and the cell pellet was stored at -80˚C. Cell pellets were thawed at 4˚C and lysed in 3 volumes of Buffer A supplemented with 1 mM DTT and protease inhibitors by glass beads beating. Whole cell lysate was spun down at 20,000 ×$g$ for 30 min. The clarified supernatant was incubated with 1 mL of anti-Flag M2 agarose beads (Sigma) at 4˚C overnight. The anti-Flag M2 beads were then washed with 20 mL Buffer A and eluted by incubating with 5 mL Buffer A with 0.25 mg/mL 3xFlag peptide with protease inhibitors at 4˚C for 1 hour. After collecting the first elution, 2 mL Buffer A was used to rinse the resins and combined with the first elution. The eluted sample was concentrated to ~0.5 mL by ultrafiltration and then fractionated on a 24 mL Superdex 200 10/300 GL column (GE Healthcare) equilibrated with 25 mM HEPES-KOH pH 7.6, 0.1 M KOAc, 0.02% NP-40, and 10% glycerol. The peak fractions were pooled, concentrated and stored in aliquots at -80˚C.

## Cdc6 and ORC purifications

Cdc6 was purified as previously described [41,51]. Rosetta cells expressing GST-Cdc6 were grown in 1 L of LB media to an OD of 0.6 at 37˚C. Cdc6 expression was induced with 0.5 mM IPTG at 18˚C for 5 hours. The cells were collected and washed in Buffer G (50 mM potassium phosphate buffer pH7.5, 5 mM MgCl$_2$, 1% Triton X-100, 1 mM DTT). The cell pellet was resuspended in 25 mL Buffer G supplemented with 2 mM ATP, 0.15 M KOAc and protease inhibitors, and lysed by sonication. The lysate was centrifuged at 20,000 ×$g$ for 30 minutes. The clarified lysate was bound to 1 mL pre-washed Glutathione Sepharose resins at 4˚C for 3 hours with rotation. After discarding the unbound fraction, the resin was washed with Buffer G with 0.15 M KOAc and 2 mM ATP. Cdc6 were eluted with 5 mL Buffer G with 0.15 M KOAc, 2 mM ATP and 15% glycerol containing 10 mM glutathione, and then treated with 100 Units preScission protease at 4˚C overnight. The purified Cdc6 protein was concentrated and stored in aliquots at -80˚C. The ORC protein was expressed and purified according to the published method with no modification [44].

## *In vitro* MCM-DH loading assay

A 2.7 kb ARS-containing DNA fragment was PCR amplified from an ARS-containing plasmid using 5' biotinylated primers (5'-CAGGAAGGCAAAATGCCGC and 5'-CACTCAACCCT ATCTCGG). The PCR products were purified after gel extraction and bound to Streptavidin M-280 Dynabeads (Invitrogen). For each MCM-DH loading reaction, 50 nM Cdc6, 50 nM ORC, and 250 nM MCM-Cdt1 were incubated with Streptavidin beads containing 1 pmol biotinylated DNA in the loading buffer (25 mM Hepes-KOH, pH 7.6, 10 mM Mg(OAc)$_2$, 20 mM KOAc, 0.02% NP-40, 2% glycerol) with 1 mM DTT, protease inhibitors and 5 mM ATP. The reactions were performed at 30˚C for 30 min with gentle shaking. After incubation, the unbound fractions were removed, and the beads were washed sequentially with 0.4 mL of ice-cold washing buffer (45 mM Hepes-KOH, pH 7.6, 5 mM Mg(OAc)$_2$, 0.02% NP-40, 1 mM

EDTA, 1 mM EGTA, 10% glycerol), ice-cold low-salt buffer (washing buffer with 0.3 M KOAc), and finally ice-cold high salt buffer (washing buffer with 0.5 M NaCl). 25 Units of Benzonase (Sigma) were added to digest the DNA in 15 μL loading buffer for 10 min at 30°C. The eluted samples were mixed with 5 μL of 4xLDS buffer (Invitrogen), heated at 95°C for 2 min and then subjected to SDS-PAGE and Silver staining.

### GCR assay

Fluctuation analysis was performed using at least 16 independent colonies per mutant isolate. The 95% confidence intervals were calculated to obtain the upper and lower limits according to the published method [52].

### Psf2 pull-down to detect CMG

Wild-type and *mcm3-2KR* strains expressing chromosomal Psf2-TAF were used to detect the CMG via anti-Flag immunoprecipitation. To ensure proper cell cycle arrest and release of the slow-growing *mcm3-2KR* cells, cells were refreshed 2 days prior to the experiment. Starting from $OD_{600}$ ~0.3 in YPD, wild-type Mcm2 and *mcm3-2KR* cells were arrested by 15 nM alpha-factor for 3 and 5 hours, respectively. After G1-arrest, ~80 OD*mL cells were collected by centrifugation. The remaining cells were released into pre-warmed YPD media and collected every 15 min for up to 3 hours after release. Cells collected at each time point were washed twice in 2 mL ice-cold washing buffer containing PBS, 5 mM EDTA, protease inhibitors, 1 mM PMSF, 5 mM NaF, 5 mM beta-glycerophosphate and 10% glycerol, and the cell pellets were stored in -80°C until use. After confirming the cell cycle arrest and release with FACS, cells were lysed in 0.8 mL ice-cold Lysis buffer (washing buffer + 0.2% NP-40 but without glycerol) using glass beads beating at 4°C. Cell extracts were centrifuged at 13,000 RPM for 10 minutes and the clarified extracts were normalized using the Bradford assay. Then, 20 μL anti-Flag M2 beads were used to incubate with ~ 730 μL of clarified extracts at 4°C for 3 hours and then each sample was washed by the ice-cold lysis buffer, Finally, the bound proteins were eluted with 60 μL lysis buffer containing 0.2 mg/mL 3xFlag peptide for 2 hours at room temperature. Both input and eluted samples were prepared for Western blot analysis using the indicated antibodies.

### Pulsed-field gel electrophoresis (PFGE)

To collect cells for PFGE analysis, wild-type and *mcm3-2KR* cells were arrested in G1 phase with 15 nM alpha factor for 3 and 6 hours, respectively. Cells were then washed with warm YPD twice to release into cell cycle. Approximately 630 million cells were collected for each time point and fixed with a final concentration of 50% ethanol and 25 mM EDTA overnight at -20°C. The next day, cell pellets were washed three times with 50 mM EDTA, resuspended with 500 μL of 0.5 M EDTA and kept in a 1.7 mL Eppendorf tube at 55°C. Plugs were made by mixing 400 μL BioRad Low-Melting Point Agarose (1% agarose dissolved in 124 mM EDTA) and 85 μL Solution I (2.55 mL SCE solution, 5.0% β-mercaptoethanol (BME), 1 mg/mL Zymolase; SCE solution consists of 1 M sorbitol, 60 mM EDTA, 100 mM sodium citrate) with the cell suspensions. After the samples were solidified in plug molds, the plugs were incubated in Solution II (450 mM EDTA, 10 mM Tris pH 8.0, 7.5% BME, 10 μg/mL RNase A) at 37°C for 1 hour. Solution II was drained, and Solution III was added (450 mM EDTA, 10 mM Tris pH 8.0, 1% Sarkosyl, 1 mg/mL Proteinase K). The plugs were subsequently incubated at 55°C for overnight. After Solution III was drained, the plugs were equilibrated with the storage buffer (50 mM EDTA and 40% glycerol) at room temperature for 30 min, washed by the storage buffer and then stored at -20°C. To analyze samples, a 1% gel (15x10cm) was made with

BioRad Pulse-Field Certified Agarose in 0.5x TBE buffer, and plugs were pipetted into the wells after being melted at 65˚C. The gel was equilibrated with 0.5x TBE at 4˚C for 30min before being placed into a BioRad CHEF 275BR chamber filled with 0.5x TBE. This chamber is connected to a BioRad CHEF-DR II Control Module 961BR, a BioRad 260BR variable pump, and a cooler. The cooler was set to 14˚C and the PFGE was run at 6.0V/cm for 26 hours with switch times from 60s to 120s. The gel was subsequently stained with Ethidium bromide before imaging.

## Flow cytometry

At each time point, 300 μL cells were collected and mixed with 700 μL 100% ethanol for fixation. The samples were then treated with 1 mL of 0.25 mg/mL RNase A and 1 mg/mL Proteinase K in 50 mM sodium citrate overnight at 37˚C. The following day, the samples were spun down and sonicated by a BRANSON Sonifier-450 for 3 seconds at the output level 5 and 100% duty cycle in 1 mL 50 mM sodium citrate with 1 μM Sytox Green. After sonication, the samples were left in the dark at room temperature for one hour. Finally, samples were analyzed using a BD LSR II Flow Cytometer and the results were processed using FlowJo v10.6.

## Supporting information

**S1 Fig.** (A) Effects of K352R, K357R, K355R/K356R and 4KR (all four lysines in this region are mutated to arginine) on sumoylated Mcm3. Total sumoylated proteins were purified by Ni-NTA and anti-Flag purifications and analyzed by anti-Mcm3 antibody. (B) FACS shows the cell cycle profile of the *mcm3-29KR-1 (K767R)* cells used in the experiment described in Fig 2E. Cells were arrested in G1 by alpha factor and G2/M by nocodazole. S phase cells were collected at 30 min after G1 release. (C) FACS shows the G1-arrested *CDC6* and *cdc6*-1 in the *mcm3-29KR-1 (K767R)* strain background. Logarithmic cells were shifted to 37˚C for 4 hours in the presence of 15 nM alpha-factor. These cells were used for the experiment in Fig 2F. (EPS)

**S2 Fig.** (A) Effects of *mcm3-2KR* on the association among MCM subunits. Flag-Mcm3 or Flag-Mcm3-2KR from G1-arrested cells were immuno-purified by anti-Flag affinity resins and probed by various anti-MCM antibodies. Lower amounts of Mcm2, Mcm6 and Mcm7 were found to associate with Flag-Mcm3-2KR than WT Flag-Mcm3, despite a higher amount of Flag-*mcm3-2KR* lysate was used. (B) G1-arrested HA-Mcm3 haploids expressing extragenic WT Flag-Mcm3 or Flag-Mcm3-2KR were used for anti-Flag immunoprecipitation. Co-purified HA-Mcm3 was detected by anti-HA Western blotting and quantified by densitometry using ImageJ. The numbers below show the relative intensity of HA-Mcm3 to that co-purified with WT Flag-Mcm3 in two experimental replicates. (C) Diploid cells expressing Flag or HA tagged Mcm3, WT or 2KR, were used for anti-Flag immunoprecipitation assay. Co-purified HA-Mcm3 was detected by anti-HA Western blotting and quantified by densitometry using ImageJ. Band intensity relative to WT Flag-Mcm3/HA-Mcm3 is shown in two experimental replicates. In each case, *mcm3-2KR* causes a reproducible reduction of MCM complex across multiple experiments. (EPS)

**S1 Table. Duplication-mediated GCR rates of the *mcm3-KR* mutants, along with the previous results of SUMO E3 ligase mutants used for comparison.** (XLSX)

**S2 Table. Yeast strains used.** (DOCX)

**S3 Table. Plasmids used.**
(DOCX)

## Acknowledgments

We thank Drs. Christopher Putnam, Kevin Corbett, Richard Kolodner and Stephen Hinshaw for critical reading of the manuscript, Drs. John Diffley (Crick Institute) and Stephen Bell (MIT) for sharing plasmids and yeast strains, Dr. Marco Foiani (IFOM) for antibodies recognizing total Rad53 and hyper-phosphorylated Rad53, and members of the Zhou laboratory for invaluable technical assistance.

## Author Contributions

**Conceptualization:** Yun Quan, Huilin Zhou.

**Data curation:** Yun Quan, Qian-yi Zhang, Huilin Zhou.

**Formal analysis:** Yun Quan, Huilin Zhou.

**Funding acquisition:** Huilin Zhou.

**Investigation:** Yun Quan, Qian-yi Zhang, Ann L. Zhou, Yuhao Wang, Jiaxi Cai, Yong-qi Gao, Huilin Zhou.

**Methodology:** Yun Quan, Qian-yi Zhang, Ann L. Zhou, Huilin Zhou.

**Project administration:** Huilin Zhou.

**Resources:** Huilin Zhou.

**Supervision:** Yun Quan, Huilin Zhou.

**Validation:** Yun Quan, Ann L. Zhou, Huilin Zhou.

**Writing – original draft:** Huilin Zhou.

**Writing – review & editing:** Yun Quan, Ann L. Zhou, Huilin Zhou.

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
