## [Decision Letter · Decision Letter 0]

14 Mar 2022

Dear Huilin,

Thank you very much for submitting your Research Article entitled 'Site-specific MCM sumoylation prevents genome rearrangements through regulated MCM loading' to PLOS Genetics.

The manuscript was fully evaluated at the editorial level and by independent peer reviewers. The reviewers appreciated the attention to an important problem and noted the high quality of the data presented, but raised some concerns about the current manuscript. In particular, it would be important to ensure that the mcm3-2KR forms the Mcm2-7 complex as well as WT, and to address a possible defect in CMG unloading. Reviewer 1 suggested that cytology would be a nice addition to support the conclusion of incomplete replication, but noted this is not essential. Based on the reviews, we will not be able to accept this version of the manuscript, but we would be willing to review a much-revised version. We cannot, of course, promise publication at that time.

If you decide to revise the manuscript for further consideration at PLOS Genetics, please aim to resubmit within the next 60 days, unless it will take extra time to address the concerns of the reviewers, in which case we would appreciate an expected resubmission date by email to plosgenetics@plos.org.

[LINK]

We are sorry that we cannot be more positive about your manuscript at this stage. Please do not hesitate to contact us if you have any concerns or questions.

Yours sincerely,

Lorraine S. Symington

Associate Editor

PLOS Genetics

Gregory P. Copenhaver

Editor-in-Chief

PLOS Genetics

Reviewer's Responses to Questions

**Comments to the Authors:**

Reviewer #1: Site-specific MCM sumoylation prevents genome rearrangements through regulated MCM loading

Yun Quan; Qian-yi Zhang; Ann L Zhou; yuhao Wang; Jiaxi Cai; Yong-qi Gao; Huilin Zhou

Regulation of the loading of the MCM complex is a key aspect to control DNA replication in eukaryotes. The authors identified two residues on Mcm3, K767 and K768, that are sumoylated and whose sumoylation is important to suppress genome instability. The authors show that mutant Mcm3 that cannot be sumoylated on these residues affects MCM complex loading onto chromatin in vivo. The complexity of SUMO modifications makes the determination of the function of individual sumoylated residues difficult. Through a tour de force multi-mutant analysis, the authors distill these two sites and provide solid evidence that they are sumoylated and functionally important. The in vitro experiments represent a nice control that the SUMO site mutants are not compromised proteins.

Overall, the data support the conclusions. For Figures 4 and 5, it is not clear if the more observed effects are reproducible and the differences significant, which requires quantification and in case of Figure 4 additional experiments to show n=3. Cytological experiments might be useful to bolster the conclusion that unfinished replication is the cause for the increase in dGCR. Some clarifications and discussion could be added in several instances, which require only text changes.

Major comments:

1) Figure 4A, B and C require quantitation. Is the effect reproducible and significant? N=3 is required.

2) Figure 5B, C require quantitation. Are the effects significant and reproducible? Fig. 5 B and C make the same point, which may obviate the need for n=3.

3) Figure 6: K768R and 2KR have nearly the same increase in dGCR, but only 3KR shows growth delay. Why? Some discussion should be added.

4) The authors hypothesize that incomplete replication is the cause for the increase in dGCR, and the results in Figure 6A are consistent with this interpretation. Are there other signs of unfinished replication, such as anaphase bridges or lagging chromosomes?

Other Points

5) Lines 12, 13: Duplicative use of support/supported.

6) Lines 34 ff: This description is rather cryptic and could be elaborated.

7) Line 86: The allele designation mcm3-29KR is used twice to denote different mutations. The nomenclature should be unique and defined.

8) Figure 1D: Nomenclature for 39KR with K767 or 768 restored is difficult to understand. Please change or explain better.

9) The potential impact of the K767/768 sites on sumoylation on other residues should be discussed more.

Reviewer #2: The MCM2-7 complex is a replicative helicase that plays a central role in eukaryotic DNA replication. SUMOylation of the MCM subunits has previously been reported (PMID 26854664). This modification occurs after the DH-MCM loading in yeast and controls the phosphorylation levels of MCMs that promotes replication initiation. In this manuscript, Quan et al. identified two lysins in yeast MCM3 (K767, 768) that were modified by SUMO and proposed that the SUMOlyation of these two resides play a role in the MCM-DH loading (in other words, "replication licensing").

Initially, the authors screened MCM3 mutants containing multiple K-to-R mutations that affected cell growth and identified K767 and K768 (Figure 1). Subsequently, they found the SUMOylation of K767 and K768 occurred in the G1 phase in a Cdc6-dependent manner, suggesting that these MCM3 modifications might be linked to the MCM loading (Figure 2). The MCM3 2KR mutant having the K767,768R mutations in the endogenous MCM3 locus also showed slow growth, and the MCM2-7 complex bound at replication origins in the G1 phase was significantly reduced (Figure 3). The MCM-DH formation in vivo, an indication of replication licensing, was also defective. However, in vitro assay for testing the MCM loading to an ARS containing DNA did not find a difference between MCM3 WT and 2KR (Figure 4). The yeast mutant expressing MCM3 2KR showed slow S phase progression and replisome formation (Figure 5). Consistently, gross chromosome replication was slow in the MCM3 2KR mutant, and they showed higher GCR (Figure 6).

Overall, the experiments are well designed and conducted. I agree with most interpretations. However, I have one major point that the authors should clarify to support their conclusion that the SUMOylation of MCM3 K767 and K768 plays a role in the MCM loading.

For phenotypic analyses in Figures 3 to 5, the authors used the MCM3 2KR mutant in which MCM3 K767,768R was expressed from the endogenous locus. I wonder if the MCM3 2KR forms the MCM2-7 complex efficiently as MCM3 WT. I understand that the mutant forms such complex when overexpressed for in vitro assay (Figure 4C). But this has not been tested in the MCM3 2KR mutant. If the MCM2-7 hexamer level was reduced in this mutant, it explains all phenotypes observed in Figures 3 to 5, which does not support the authors' proposal. Therefore, the authors should compare the level of the MCM2-7 complex between the MCM3 WT and 2KR cell lines.

Minor issues

Lines 117-118: "Thus, mcm3-2KR cells have an origin independent MCM-DH loading defect."

What does it mean? Should it be "Thus, mcm3-2KR cells have an MCM loading defect at replication origins"?

Supplementary Table 2: The genotype of HZY2337 and HZY1954 appears to be identical.

Reviewer #3: The manuscript entitled ”Site-specific MCM sumoylation prevents genome rearrangements through regulated MCM loading” by Quan and colleagues reports that MCM3 is sumoylated at several residues in a redundant fashion by Siz1, Siz2 and/or Mms21 and that the sumoylation at K767/K768 is critical for timely DNA replication and suppression of GCRs. An mcm3-K767R-K768R mutant exhibits reduced binding to origins and a lower level of MCM double-hexamer formation in vivo but no defect in DNA binding in vitro. These findings suggest that Mcm3 sumoylation is important for regulating some aspect of MCM function in vivo. These findings are novel and interesting and the data is of high quality, but the mechanism that is regulated by sumoylation is not identified. However, a few clarifying experiments could help narrow down the mechanism and make the manuscript suitable for publication in PLoS Genetics.

Major issues:

1. Line 113-118 and title: The authors suggest that the mcm3-2KR mutant impairs MCM loading at origins. It may equally be that sumoylation protects MCM against untimely unloading. The authors should provide experimental data to distinguish between these two models.

2. Is it the CMGs that contain sumoylated MCMs that fire replication in S phase?

3. What fraction of MCM3 is sumoylated at K767/K768? The observation that the three major MCM3-SUMO bands can be removed independently by single mutations and no higher order species are observed suggest that MCM3 is primarily mono-sumoylated at only 1 lysine, which indicates that either sumoylation at the three sites are mutually exclusive or more likely that only a small fraction of MCM3 is sumoylated at any given time. Knowing the fraction of sumoylated Mcm3 is important because it would help distinguish if SUMO serves a constitutive function in MCM or a transient role e.g. during opening of the DNA gate in MCM to load MCM.

4. Line 175 and elsewhere: There is good evidence that MCM is not loaded as a double hexamer, but rather that single hexamers are loaded sequentially to form the double hexamer (Miller et al, 2019). Therefore, the reduced level of MCM-DH likely reflects a reduction in DNA-bound MCM and therefore less opportunity for the MCM-DH to form. Please modify the text accordingly.

5. The authors examine MCM loading at an early and late ARS, but the notion that MCM3 sumoylation promotes genome-wide loading onto DNA would be strengthen, if the authors could demonstrate that mcm3-2KR locates to the soluble fraction and MCM3 to the chromatin fraction in WCE.

Minor corrections

1. In the introduction on page 4, the impression is given that the regulation of replication by SUMO is ”not supported by direct evidence”, which does not give sufficient credit to the extensive work on PCNA sumoylation in both yeast and human.

2. Figure 1D: the labels “K768R” are confusing because these cells are K768 WT, if I understand the figure correctly.

3. Line 157: change “Pulse-field gel electrophoresis” to “Pulsed-field gel electrophoresis”.

References:

Miller TCR, Locke J, Greiwe JF, Diffley JFX, Costa A (2019) Mechanism of head-to-head MCM double-hexamer formation revealed by cryo-EM. Nature (London) 575: 704-710

**Have all data underlying the figures and results presented in the manuscript been provided?**

Reviewer #1: Yes

Reviewer #2: Yes

Reviewer #3: Yes

PLOS authors have the option to publish the peer review history of their article (what does this mean?). If published, this will include your full peer review and any attached files.

Reviewer #1: No

Reviewer #2: No

Reviewer #3: No

---

## [Decision Letter · Decision Letter 1]

22 May 2022

Dear Huilin,

Thank you very much for submitting your Research Article entitled 'Site-specific MCM sumoylation prevents genome rearrangements by controlling origin-bound MCM' to PLOS Genetics.

The manuscript was fully evaluated at the editorial level and by independent peer reviewers. Reviewer 1 was satisfied with the revised version, but Reviewers 2 and 3 both expressed concern over the mechanism proposed for Mcm3 K767/768 sumoylation in regulation of MCM loading.

We therefore ask you to modify the manuscript according to the review recommendations. Your revisions should address the specific points made by each reviewer.

[LINK]

Yours sincerely,

Lorraine S. Symington

Associate Editor

PLOS Genetics

Gregory P. Copenhaver

Editor-in-Chief

PLOS Genetics

Reviewer's Responses to Questions

**Comments to the Authors:**

Reviewer #1: Site-specific MCM sumoylation prevents genome rearrangements through regulated MCM loading

Yun Quan; Qian-yi Zhang; Ann L Zhou; Yuhao Wang; Jiaxi Cai; Yong-qi Gao; Huilin Zhou

The revisions in the re-submitted manuscript address my comments and concerns, strengthening an already interesting study.

Reviewer #2: Thanks for showing the MCM3 IP data to compare the MCM2–7 hexamer levels in WT and 2KR cells (Figures 4A and S2A). The data indicated that the MCM3-2KR mutant formed the hexamer less efficiently in the G1 phase. I have some comments on the authors' interpretation and would like to discuss the following issues.

The authors interpreted that they mainly detected "loaded MCMs" in the G1 phase (lines 124 to 128). However, I cannot agree with your interpretation.

If the authors detected only loaded MCMs, all subunits in WT and 2KR should be proportional because the loaded MCM2–7 complex must be hexameric. Therefore, the authors likely detected both "loaded and pre-loaded MCMs" in the G1 phase (Figures 4A and S2A). If the authors wished to detect only pre-loaded MCMs, they could have used a GAL-CDC6 background to shut off Cdc6 expression. But the current data already suggested that the functional MCM2–7 hexamers before DNA loading were reduced in the MCM3-2KR mutant to some extent. As I pointed out previously, reduced levels of the functional MCM2–7 hexamer potentially explain all defects presented in this paper. Unfortunately, the data in Figures 4A and S2A do not exclude this possibility.

Another issue is that the authors wrote "MCM loading occurs sequentially, first as MCM single hexamer (MCM-SH) and then MCM double hexamer (MCM-DH) (lines 123 to 124)" as if the two forms can be seen in vivo. MCM-SH on DNA (also known as OCCM) is an intermediate complex that can only be detected in the presence of ATP-gS in vitro (PMIDs 23474987 and 23603117). Moreover, a recent paper showed a mechanism of coupled MCM loading for forming MCM-DH (PMID 31748745). As far as I know, stable MCM-SH on DNA has not been detected in vivo, and all loaded MCMs are MCM-DH.

Therefore, I suggest discussing the possibility of reduced expression of the functional MCM2–7 hexamer in MCM3-2KR and tone down the hypothesis that the SUMO modification of MCM3 promotes MCM loading. I also suggest correcting the description of MCM-SH and MCM-DH in the text.

Minor point

The experimental procedures of the MCM3 IP (Figures 4A and S2A) are not described in Materials & Methods.

Reviewer #3: The manuscript entitled ”Site-specific MCM sumoylation prevents genome rearrangements by controlling origin-bound MCM” by Quan and colleagues has been extensively revised and rewritten. This has improved the manuscript, but unfortunately the manuscript still does not provide much insight into how sumoylation might regulate the steady-state level of loaded MCM on chromatin in G1.

In response to the reviewers comments to the original manuscript, the authors replied “The fraction of Mcm3 that is sumoylated at K767/768 in cells is low”, but this information is not documented and incorporated into the manuscript, which it should be. An approximation of the fraction of sumoylated MCM would make it easier to develop and test models for the biological function of Mcm3 that is sumoylated at K767/768.

The reviewers other questions were addressed.

**Have all data underlying the figures and results presented in the manuscript been provided?**

Reviewer #1: Yes

Reviewer #2: Yes

Reviewer #3: Yes

PLOS authors have the option to publish the peer review history of their article (what does this mean?). If published, this will include your full peer review and any attached files.

Reviewer #1: No

Reviewer #2: No

Reviewer #3: No

---

## [Editor Report · Decision Letter 2]

25 May 2022

Dear Huilin,

We are pleased to inform you that your manuscript entitled "Site-specific MCM sumoylation prevents genome rearrangements by controlling origin-bound MCM" has been editorially accepted for publication in PLOS Genetics. Congratulations!

Yours sincerely,

Lorraine S. Symington

Associate Editor

PLOS Genetics

Gregory P. Copenhaver

Editor-in-Chief

PLOS Genetics

Comments from the reviewers (if applicable):

**Data Deposition**

http://datadryad.org/submit?journalID=pgenetics&manu=PGENETICS-D-22-00170R2

**Press Queries**

---

## [Editor Report · Acceptance letter]

7 Jun 2022

PGENETICS-D-22-00170R2 

Site-specific MCM sumoylation prevents genome rearrangements by controlling origin-bound MCM 

Dear Dr Zhou, 

We are pleased to inform you that your manuscript entitled "Site-specific MCM sumoylation prevents genome rearrangements by controlling origin-bound MCM" has been formally accepted for publication in PLOS Genetics! Your manuscript is now with our production department and you will be notified of the publication date in due course.

With kind regards,

Zsofia Freund

PLOS Genetics

On behalf of:
